# Simultaneous Recognition and Detection of Adenosine Phosphates by Machine Learning Analysis for Surface-Enhanced Raman Scattering Spectral Data

**DOI:** 10.3390/s24206648

**Published:** 2024-10-15

**Authors:** Ryosuke Nishitsuji, Tomoharu Nakashima, Hideaki Hisamoto, Tatsuro Endo

**Affiliations:** 1Department of Information Networking, Graduate School of Information Science and Technology, Osaka University, 2-8 Yamadaoka, Suita 565-0871, Osaka, Japan; nishitsuji@ist.osaka-u.ac.jp; 2Department of Interdisciplinary Informatics, Graduate School of Informatics, Osaka Metropolitan University, 1-1 Gakuencho, Nakaku, Sakai 599-8531, Osaka, Japan; tomoharu.nakashima@omu.ac.jp; 3Department of Applied Chemistry, Osaka Metropolitan University, 1-1 Gakuencho, Nakaku, Sakai 599-8531, Osaka, Japan; hisamoto@omu.ac.jp

**Keywords:** surface-enhanced Raman scattering, gold nanostructure, adenosine phosphates, machine learning, small data analytics

## Abstract

Adenosine phosphates (adenosine 5′-monophosphate (AMP), adenosine 5′-diphosphate (ADP), and adenosine 5′-triphosphate (ATP)) play important roles in energy storage and signal transduction in the human body. Thus, a measurement method that simultaneously recognizes and detects adenosine phosphates is necessary to gain insight into complex energy-relevant biological processes. Surface-enhanced Raman scattering (SERS) is a powerful technique for this purpose. However, the similarities in size, charge, and structure of adenosine phosphates (APs) make their simultaneous recognition and detection difficult. Although approaches that combine SERS and machine learning have been studied, they require massive quantities of training data. In this study, limited AP spectral data were obtained using fabricated gold nanostructures for SERS measurements. The training data were created by feature selection and data augmentation after preprocessing the small amount of acquired spectral data. The performances of several machine learning models trained on these generated training data were compared. Multilayer perceptron model successfully detected the presence of AMP, ADP, and ATP with an accuracy of 0.914. Consequently, this study establishes a new measurement system that enables the highly accurate recognition and detection of adenosine phosphates from limited SERS spectral data.

## 1. Introduction

The term adenosine phosphates (APs) refers to adenosine 5′-monophosphate (AMP), adenosine 5′-diphosphate (ADP), and adenosine 5′-triphosphate (ATP), which pertain to energy storage and signaling between cells and play important physiological roles [1,2]. In addition, APs have attracted attention as non-invasive diagnostic and prognostic biomarkers for diseases such as Alzheimer’s and Parkinson’s [3,4,5]. Since APs are in equilibrium (2ADP↔ATP+AMP) via adenylate kinases in phosphotransfer systems, AP ratios are important physiological factors. Therefore, their simultaneous detection is necessary to acquire insight into these complex physiological energy processes. Currently, measurement using the luciferase reaction is the main method for detecting APs [6,7]. However, the information obtained from this measurement is limited and does not provide detailed information on the ratio of APs. Moreover, as APs are very similar in terms of molecular size, charge, and structure, their simultaneous detection is challenging. Although, high-performance liquid chromatography (HPLC) methods have been studied for the simultaneous detection of APs [8,9], these methods require professional knowledge and operator skills for the pretreatment of analytes and complex measurement operations.

In recent years, surface-enhanced Raman scattering (SERS) has attracted considerable attention as a highly sensitive and selective measurement method. SERS is an enhancement of the Raman scattering intensity of molecules adsorbed on metallic nanostructures such as nanoparticles [10,11], nanorods [12,13] and nanowires [14,15]. SERS enables the detection of Raman scattering from molecules, which was previously difficult to detect because of the weak intensity. SERS provides unique spectroscopic fingerprints of chemicals and biomolecules through molecular vibrations, thereby allowing the identification of numerous analytes with high discrimination [16,17]. Therefore, SERS has been studied for applications in various fields in environmental [18], food [19], materials [20], and bioanalysis [21,22]. However, similar molecules are difficult to identify from their spectra because SERS produces spectra of similar shapes from molecules with similar properties and structures [23]. Therefore, the simultaneous recognition and detection of APs using SERS is challenging. Although studies determining the ATP:ADP ratio of SERS spectra have been reported [24], there are no reports showing that AMP, ADP, and ATP can be recognized and detected simultaneously using SERS.

Recent studies have increasingly employed machine learning techniques to discriminate analytes based on subtle variations in their spectral shapes. Machine learning has been applied to spectral analysis using mass spectrometry [25], nuclear magnetic resonance [26], ultraviolet-visible spectroscopy [27], near-infrared spectroscopy [28], and Raman spectroscopy [29]. Because spectral data contain considerable chemical information about the analytes, conventional univariate analysis ignores substantial information. Therefore, spectral analysis using machine learning, which is a multivariate analysis method, has attracted considerable attention in chemistry [30]. Various molecules have been identified using machine learning to extract chemical information from spectral data. For example, SERS spectra were analyzed by machine learning to identify the types of bacteria and viruses [31,32]. Different types of bacteria and viruses exhibit distinct structures, and these differences are reflected in the shape of the SERS spectrum. Machine learning models can determine the types of bacteria or virus by recognizing differences in the shapes of those SERS spectra. A large amount of training data is essential for a machine learning model to recognize the differences in SERS spectral shapes. However, the amount of data available in the field of chemistry is generally limited. Therefore, analytical methods for small amounts of SERS spectral data are required.

In this study, to recognize and detect the types of APs (AMP, ADP, and ATP) in a mixed solution with high accuracy, a machine learning model was trained on SERS spectra using feature selection and data augmentation (Figure 1). First, we fabricated a SERS substrate with a gold nanodisk array (GNDA) on which nanoparticles were adsorbed to obtain the SERS spectra of the APs. The SERS spectra of the mixed AP solutions were measured using these SERS substrates. Subsequently, for the numerous noisy regions in the measured SERS spectra that contained no chemical information, feature selection was implemented using k-means to extract the regions that contained chemical information. Noise was added and the synthetic minority oversampling technique (SMOTE) was applied for data augmentation training. These processes improved the prediction accuracy of each machine learning model, in the case of multilayer perceptron (MLP) by 0.100, reaching an accuracy of 0.914. These results suggest that with limited data, APs can be recognized and detected with high accuracy using feature selection and data augmentation processing.

## 2. Materials and Methods

### 2.1. Fabrication of SERS Substrates

Figure 2 shows the fabrication procedure for the SERS substrate. First, GNDAs were fabricated by depositing gold on cyclo-olefin polymer (COP)-based films (FLP230/200-120, Scivax Co., Ltd., Kanagawa, Japan) with periodic nanostructures. The COP films were cleaned using 2-propanol (Kanto Chemical Co. Inc., Tokyo, Japan) and ultrapure water, and then dried under airflow. A gold layer (thickness: 100 nm) was thermally deposited onto the COP films using a thermal evaporator (SVC-700TM/700-2; Sanyu Electron Co., Ltd., Tokyo, Japan). Second, silica nanoparticles (Polysciences, Inc., Warrington, DC, USA) were electrostatically adsorbed onto gold nanostructures using the layer-by-layer (LbL) method. For the LbL method, 3 mg/mL aqueous solutions of poly (allylamine hydrochloride) (PAH) (Sigma-Aldrich Japan Inc., Tokyo, Japan) and poly (sodium 4-styrenesulfonate) (PSS) (Sigma-Aldrich Japan Inc., Tokyo, Japan) containing NaCl (0.5 M) (Wako Pure Chemical Co., Osaka, Japan) were used. The fabricated GNDAs were immersed in a PAH solution for 1 min at room temperature (20–25 °C), followed by washing with ultrapure water. After washing, the GNDAs with the PAH layer were immersed in a PSS solution under the same conditions, followed by washing with ultrapure water. This coating process was repeated twice, and finally the surfaces were coated with the PAH layer, followed by washing with ultrapure water. GNDAs with positively charged surface were immersed in 0.25 mg/mL silica nanoparticle (ϕ30, 50, 100 nm) dispersion for 5 min at room temperature (20–25 °C). Following immersion, the GNDAs were washed with ultrapure water and dried under airflow. The SERS substrates were prepared by re-depositing a gold layer (thickness: 30 nm) onto the GNDAs.

### 2.2. SERS Measurements

Sample solutions for measurement were prepared by mixing 1 mM AMP (≧99%, Sigma-Aldrich Japan Inc., Tokyo, Japan), ADP (≧95%, Sigma-Aldrich Japan Inc., Tokyo, Japan), and ATP (≧99%, Sigma-Aldrich Japan Inc., Tokyo, Japan) in an aqueous solution in several volume ratios (1:0:0, 0:1:0, 0:0:1, 1:1:0, 1:0:1, 0:1:1, 1:1:1). The SERS substrates were dried at room temperature (20–25 °C) after placing a drop of the sample solution (5 mL) onto them to equalize the adsorption of the APs. All SERS spectra were acquired using a laser confocal Raman microscope (RAMAN-11, Nanophoton, Osaka, Japan) with a 785 nm laser. In this study, the SERS measurements were performed under the following conditions: 50× objective lens (N.A. = 0.8), 1 mW laser power, 60 s integration time, and 50 μm slit width.

### 2.3. Data Preprocessing

In spectral analysis using machine learning, data preprocessing is an important process. A Raman shift range of 200–1700 cm^−1^, which is the fingerprint region, was selected for the analysis because the fingerprint region contains a large amount of chemical information. The baselines of the SERS spectra were corrected and smoothed using the RAMAN Imager software (version 2, Nanophoton, Osaka, Japan). The SERS spectra were smoothed using the Savitzky–Golay algorithm with a second-degree polynomial and a window size of five [33,34] after correcting the baseline by fitting a quintic function. The intensity of the SERS spectra depended on the conditions of the measurement equipment and fabricated substrates. Therefore, the SERS spectra were standardized to align the intensity scale. Spectral standardization was calculated as follows:(1)$$I_{j,\mathrm{std}}=\frac{I_{j}-\bar{I}}{s}$$
where *I*_j,std_ is the standardized intensity of each Raman shift; *I*_j_ is the intensity of each Raman shift; *I* is the average intensity of the entire spectrum; and *s* is the standard deviation of the intensity of the entire spectrum. The spectrum was standardized by applying Equation (1) to the intensity of each Raman shift from 200 to 1700 cm^−1^. Because the SERS spectra contained noise regions, using the intensities of the entire spectra as feature values reduced the learning efficiency. Thus, specific Raman shifts were extracted from the SERS spectra to improve the training efficiency of the machine learning model. The Raman shifts of the measured SERS spectral data were divided into clusters using the k-means clustering algorithm. The optimal number of clusters was determined using the elbow method for k-means clustering. After clustering, significant Raman shifts for machine learning analysis were selected by removing clusters containing multiple noisy regions.

### 2.4. Data Analysis

The SERS spectra data, preprocessed as described in Section 2.3, were input into the machine learning models. A total of 140 data points were obtained for analysis in this study, with 20 points for each mixing ratio. The data were divided into 70 training data points (10 data points × 7 groups) and 70 test data points (10 data points × 7 groups). To improve the performance and robustness of the machine learning model, the amount of training data was increased by adding random noise and applying the SMOTE algorithm [35]; 2800 synthetic training data points with random noise were generated from 70 training data points, and 700 additional synthetic training data points were generated using the SMOTE algorithm. Random noise was calculated as expressed in Equation (2).
(2)$$n_{i,j}=s_{i,j}\times \;N\left(0,\;1\right)$$
where *n*_i,j_ is random noise of group i (i = 1–7) and Raman shift j (j = 200–1700 cm^−1^); *s*_i,j_ is the standard deviation of the intensity of group i and Raman shift j; and *N*(0, 1) is a value generated from a standard normal distribution. In the present study, the presence or absence of each AP was predicted using a classification framework. We selected logistic regression (LR) [36], decision tree (DT) [37], k-nearest neighbor (kNN) [38], linear discriminant analysis (LDA) [39], support vector machine (SVM) [40], random forest (RF) [41], and MLP [42] as classification frameworks. These machine learning models were implemented using scikit-learn 1.4.2 [43], which is a free software machine learning library for the Python programming language. The hyperparameters for each machine learning model were optimized via grid search. The hyperparameters and their ranges are listed in Appendix A. The other hyperparameters were adopted from the initial settings defined in scikit-learn 1.4.2. Accuracy was selected as the evaluation index for the machine learning models. All programs were run using Python 3.8. All data points were classified by swapping the training data with the test data and implementing the same process.

## 3. Results and Discussion

### 3.1. Evaluation of the Fabricated SERS Substrates

Substrates were fabricated with nanoparticles adsorbed on GNDAs (measurement substrates). The SEM images of these measurement substrates are shown in Figure 3. For comparison, substrates with nanoparticles adsorbed on gold planes (GPs) were also fabricated. The SEM images are shown in Appendix A. Figure 3 shows that the substrate was successfully fabricated with nanoparticles adsorbed onto the GNDAs, and the smaller the size of the nanoparticles, the more densely they are adsorbed. The electrostatic repulsion caused by the carboxyl groups modified on the nanoparticle surface is one factor in this result [44]. Larger nanoparticles have larger areas in contact with each other; therefore, many carboxyl groups affect the electrostatic repulsion between large nanoparticles, preventing them from being densely adsorbed. This is one factor, and a lot of others influence it. More detailed experiments are needed to identify the exact factors.

The reflection spectra of the fabricated SERS substrates are shown in Figure 4. The vertical axis in Figure 4 shows the normalized reflection intensity (N. R. I.). As shown in Figure 4a, the substrates with a GNDA structure have a broad absorption peak at approximately 785 nm, which is thought to induce SERS, whereas in Figure 4b, there is no absorption peak at approximately 785 nm (excitation light wavelength) for the substrate with nanoparticles adsorbed on the GPs. Therefore, Raman scattering was not enhanced for GPs. Although the adsorption of ϕ30 or ϕ50 nanoparticles on the GNDA structure made the absorption peak sharp (Figure 4a), the weak SERS intensity for the 785 nm excitation is possibly caused by the red shifts of the absorption peak. When the ϕ100 nm nanoparticles were adsorbed on the GNDAs, a sharp absorption peak appeared at approximately 785 nm. Therefore, the substrates with adsorbed ϕ100 nm nanoparticles display the largest SERS enhancement effect among the tested substrates. Moreover, since a wavelength of the absorption peak shifts from 785 nm when larger diameter nanoparticles are adsorbed on the GNDAs, ϕ100 nm nanoparticles are optimal.

The ATP SERS spectra were acquired using the fabricated substrate (Figure 5). The shapes of the measured SERS spectra of ATP are similar to those in a previous paper [24]. This result indicates that the SERS spectra of ATP adsorbed on the substrate were measured. Figure 5 shows that the SERS intensity differs for each substrate. The intensities of these spectra were compared for the band around 740 cm^−1^, which is the sharpest and most intense band in the spectra (Figure 6). Error bars indicate the standard deviation of band intensities. Figure 6 shows that the SERS intensity of the GNDAs is stronger than that of the gold planes. In addition, the SERS intensity is the strongest for the measurement substrate with ϕ100 nm nanoparticles adsorbed. This result is consistent with that of Figure 4 and shows that Raman scattering interacts with the localized surface plasmon resonance based on the absorption at approximately 785 nm, resulting in enhanced SERS.

To investigate the SERS intensity enhanced by the adsorption of 100 nm diameter nanoparticles, the enhanced electric field distributions of the substrates with and without ϕ100 nm nanoparticles adsorbed on the GNDAs were calculated using finite-difference time domain (FDTD) simulations (Lumelical Solutions, Inc., Vancouver, BC, Canada). In the simulation model, a 3D model of GNDAs was created and a plane wave was set as a light source. A field monitor was set to calculate the electric field and reflected light. Periodic boundary conditions were set in the x–y direction, and a perfect matching layer was set in the z direction (Appendix A). The reflection spectrum of each model was calculated and compared with the actual spectrum (Appendix A). As a result, spectra very similar to the actual measured spectra were obtained. Thus, we considered the simulation models used in the FDTD simulation to be valid for the created substrate. The simulated enhanced electric field distributions at 785 nm from these models are shown in Appendix A, where a strongly enhanced electric field is observed between the GNDAs and nanoparticles. Thus, the experimentally obtained SERS intensity is considered to be enhanced by the enhanced electric field. The fabrication method selected in this study was a simple and inexpensive method that does not require the expensive equipment used in electron beam lithography and ion etching, which are common fabrication methods for SERS substrates [45]. Based on these results, the SERS substrates were successfully fabricated using a cost-effective fabrication method.

### 3.2. Detection of APs by Analyzing SERS Spectra with Machine Learning

The SERS spectra of the samples dropped onto the SERS substrate in different mixing ratios were measured (Figure 7). Each spectrum represents an average of each mixing ratio. Figure 7 shows that the intensity and shape of the SERS spectra differ slightly depending on whether AMP, ADP, or ATP was included. The SERS spectra of APs were compared with the conventional Raman spectra of them (Appendix A). The conventional Raman spectra of APs was measured on their powder on glass substrates using the same equipment (2 mW laser power, other same conditions as SERS) as SERS. Appendix A shows that the bands at approximately 730, 1330, and 1460 cm^−1^ are selectively enhanced. These bands are related to the adenine moiety [46]. Therefore, the adenine moiety is considered to be adsorbed on the surface of SERS substrates.

Raman shift clustering was implemented using the k-means clustering algorithm for feature selection. The number of clusters was determined to be four using the elbow method (Appendix A). The clusters to which each Raman shift belongs are shown in Figure 8. Cluster 1 contains bands at approximately 1330 cm^−1^ (CN, CC stretch) [46]. Cluster 2 contains a band at approximately 730 cm^−1^ (ring breathing) and a band at approximately 1460 cm^−1^ (N7-C8 stretch, C8-H bend, NH_2_ scissor) [46]. Cluster 3 contains bands at 1075 cm^−1^ (C-O, C-O-C, C-N stretch) and 1180 cm^−1^ (P=O and C-N stretch), and other weak fingerprint regions [47]. Since Cluster 4 does not contain any characteristic bands, it was judged to be a cluster of noise regions. Based on this result, feature selection was implemented by excluding Cluster 4, which contains numerous noisy regions.

To verify the effects of feature selection and data augmentation, raw data (RD), feature selection-processed data (FSD), and feature selection and data augmentation-processed data (DAD) were input into each machine learning model, and the prediction accuracy was compared (Figure 9). Prediction accuracy was higher for FSD than for RD for most machine learning models. These results indicate that noisy regions with no chemical information inhibit the learning of machine learning models. However, only the LDA model demonstrated a reduction in prediction accuracy. This is because the LDA model predicted based on differences in noisy regions to the classification. Therefore, it is possible that LDA’s prediction for RD was inaccurate. When DAD was input into the machine learning models, many of the models improved in prediction accuracy, whereas others did not. Among the models with improved accuracy, LDA and DT improved significantly, with improvements of 0.221 and 0.093, respectively, whereas the accuracies of kNN and SVM decreased by 0.171 and 0.057, respectively. For kNN and SVM, overfitting the augmented data may decrease prediction accuracy. In other models, data augmentation improved the efficiency and robustness of machine learning. As described above, a highly accurate prediction of the existence of molecules with similar structures was achieved by feature selection and data augmentation in small data sets. In particular, it is interesting to note that the prediction accuracy of 0.900 was achieved for LR, a simple classification model, using DAD as training data. Feature selection and data augmentation are essential preprocessing techniques to enhance the accuracy of machine learning predictions when the data set is limited.

Finally, the prediction results were investigated for each machine learning model (Appendix A). These results show that the prediction of AMP:ADP:ATP = 1:1:1 is highly erroneous in most models. On the other hand, models with high prediction accuracy show relatively few errors in the prediction of that. When there are three or more molecules with similar structures, such as AP, a prediction of mixing ratio becomes more difficult. It is important to improve the accuracy of this prediction for applications. Therefore, if a more accurate or quantitative prediction is needed, data preprocessing and machine learning models must be improved.

## 4. Conclusions

In this study, APs in mixed solutions were simultaneously recognized and detected by applying machine learning analysis to SERS spectra. First, SERS substrates with nanoparticles adsorbed on GNDAs were fabricated for the SERS spectral measurement of APs. The substrate with 100 nm nanoparticles adsorbed had a sharp absorption peak at approximately 785 nm, which resulted in a stronger SERS intensity of ATP than the substrate with nanoparticles of other sizes. We succeeded in fabricating a substrate that enabled the measurement of the SERS spectra of ATP using an inexpensive simple method. Next, feature selection and data augmentation were performed to predict the presence or absence of APs by performing machine learning analysis on limited SERS spectral data. During feature selection, noise regions were excluded by clustering using the k-means algorithm. In data augmentation, the training data were increased by adding noise and using the SMOTE algorithm. A prediction accuracy of 0.914 was achieved by training the MLP using the processed data. Thus, the preprocessing enabled highly accurate simultaneous recognition and detection of APs in limited data by conducting data analysis.

## Figures and Tables

**Figure 1 sensors-24-06648-f001:**
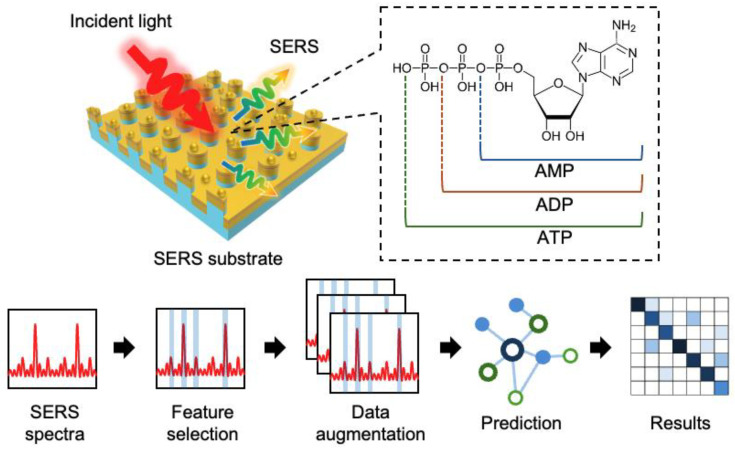
Concept of recognition and detection of APs using SERS and machine learning.

**Figure 2 sensors-24-06648-f002:**
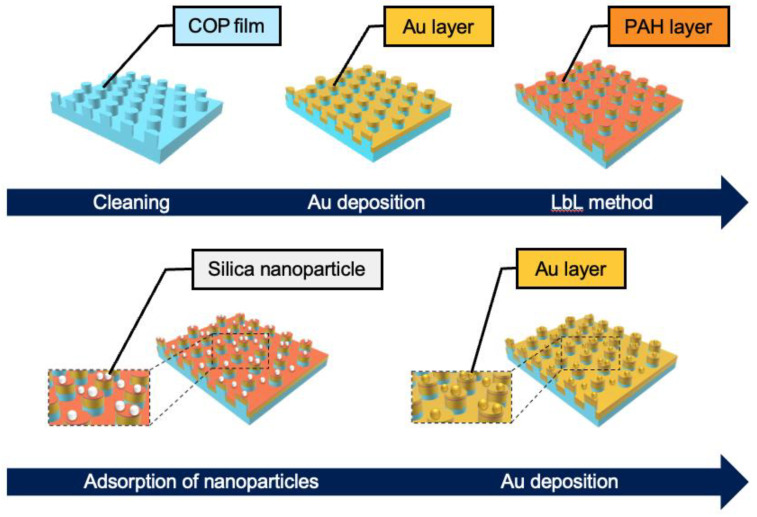
Fabrication procedure for the SERS substrate.

**Figure 3 sensors-24-06648-f003:**
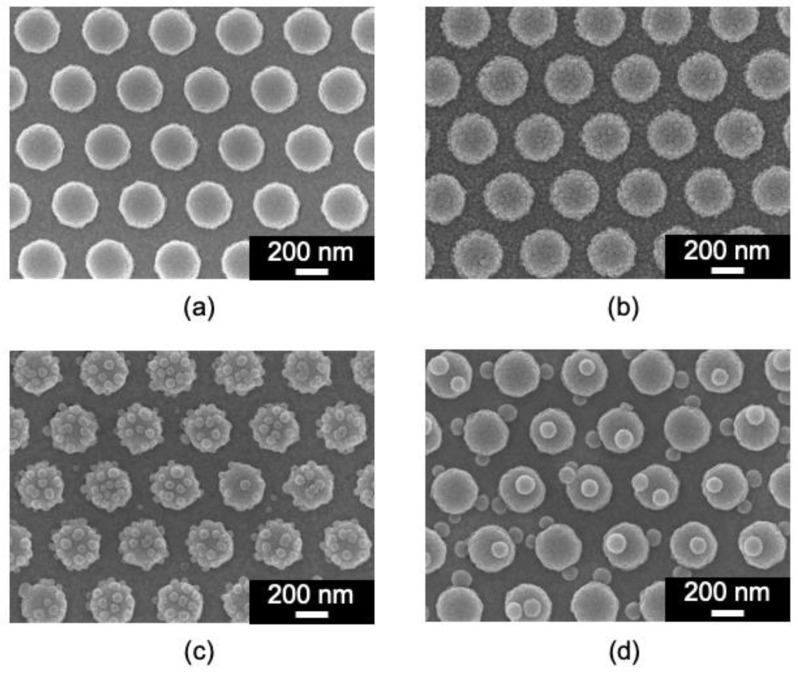
SEM images of GNDAs (**a**) without and with (**b**) 30 nm, (**c**) 50 nm, and (**d**) 100 nm nanoparticles adsorbed.

**Figure 4 sensors-24-06648-f004:**
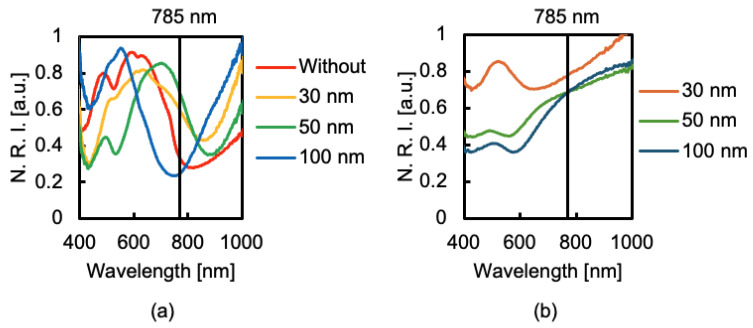
Reflection spectra of the fabricated substrates with nanoparticles adsorbed on (**a**) GNDAs and (**b**) GPs.

**Figure 5 sensors-24-06648-f005:**
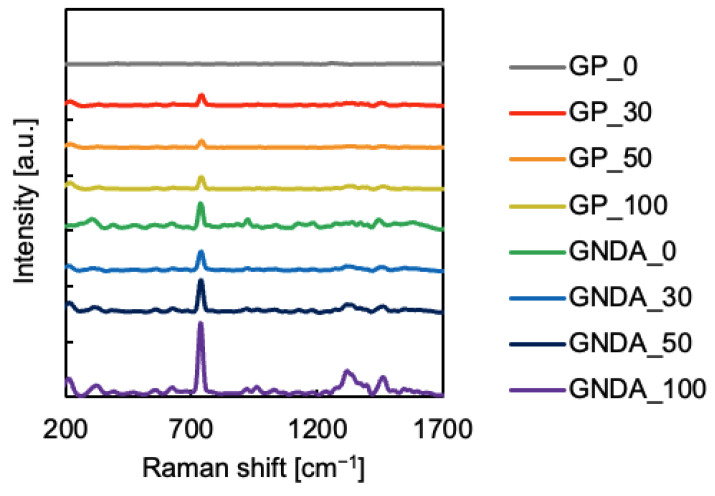
SERS spectrum of ATP measured with each substrate.

**Figure 6 sensors-24-06648-f006:**
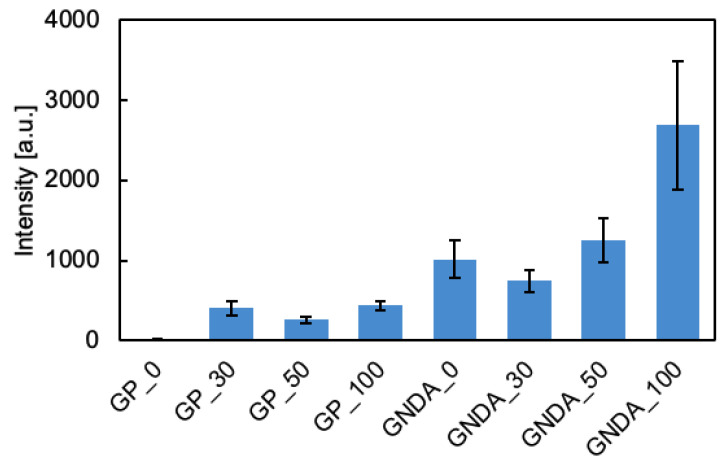
Comparison of SERS intensity on each substrate; x labels show the substrate on which the nanoparticles are adsorbed (GP or GNDA) and the size of the nanoparticles (0–100 nm).

**Figure 7 sensors-24-06648-f007:**
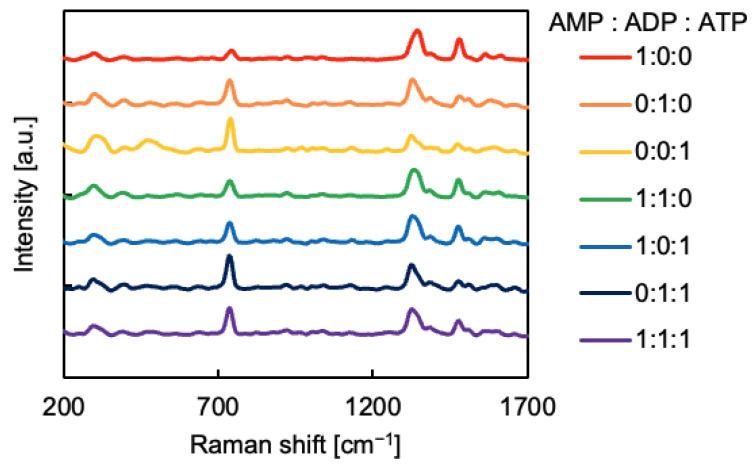
Average SERS spectrum for each mixing ratio.

**Figure 8 sensors-24-06648-f008:**
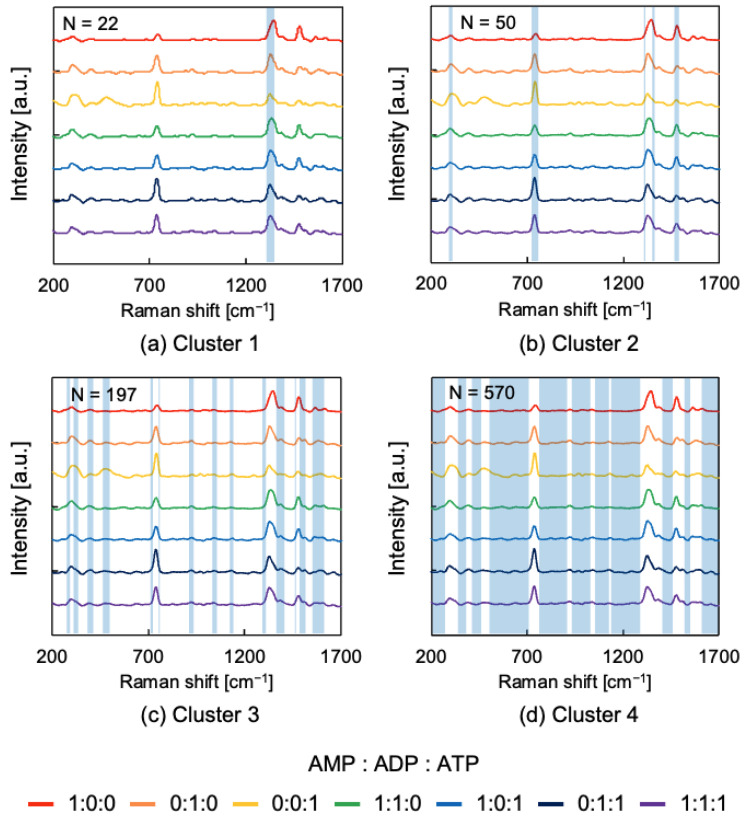
Clustering of Raman shift by k-means algorithm. N indicates the number of Raman shifts present in each cluster.

**Figure 9 sensors-24-06648-f009:**
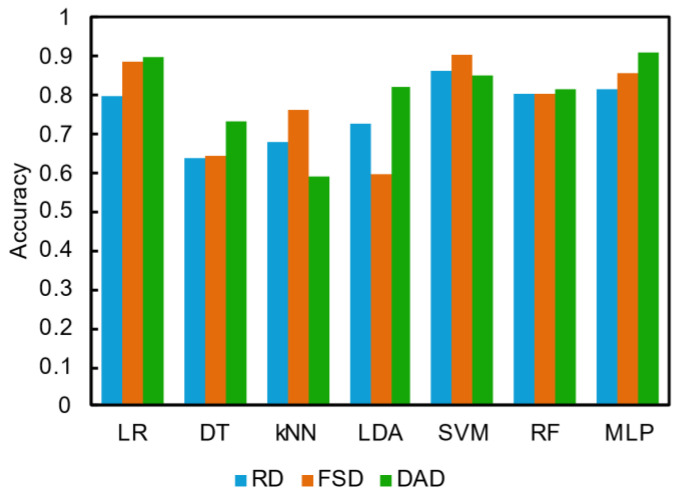
Comparison of prediction accuracy of machine learning models trained on raw (bule), feature selection-processed (orange), and data augmentation-processed (green) data.

## Data Availability

The data presented in this study are available in the article and Appendix A.

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
