# Peer review of "Simultaneous Recognition and Detection of Adenosine Phosphates by Machine Learning Analysis for Surface-Enhanced Raman Scattering Spectral Data"

_sensors, 2024, doi:10.3390/s24206648_

Round 1
Reviewer 1 Report
Comments and Suggestions for Authors
This study presents a significant advancement by demonstrating that a support vector machine model can effectively identify and detect AMP, ADP, and ATP from limited SERS data with high accuracy. By integrating SERS with machine learning, the research innovatively creates a measurement system that performs well even with limited data. However, I have some doubts, and the manuscript can be recommended for publication after the author has addressed the following poinb
1. In section 2.2 SERS Measurements, was a buffer solution used for the preparation of 1 mM AMP? Was the pH adjusted? A suitable pH and solution environment are crucial for maintaining the stability of AMP, ADP, and ATP. This section should provide more specific experimental details.
2. The statement in Line 190, "Larger nanoparticles have larger areas……from being densely adsorbed" is not entirely accurate. While it is true that larger particles have greater contact areas, which may involve more carboxyl groups participating in electrostatic repulsion, reducing the likelihood of dense adsorption, the repulsion between particles is not solely dependent on the surface contact area. It is also influenced by the uniformity of particle distribution on the substrate, adsorption forces, and solution conditions. Other factors, such as van der Waals forces and geometric constraints, may also play a role in the adsorption process of larger particles. It is recommended to supplement this with more sound physicochemical evidence.
3. This study optimized nanoparticles with diameters of F30 nm, 50 nm, and 100 nm, demonstrating that 100 nm performed best. Could larger diameter nanoparticles potentially exhibit superior performance? It is suggested to test the performance of nanoparticles with larger diameters or provide justification for why larger diameters might be unsuitable.
4. Although the elbow method determined that the number of clusters is four, whether this is truly sufficient to represent the complexity of the data should be further evaluated based on the specific characteristics of the data and experimental results. It is recommended to employ appropriate external validation methods, such as comparison with spectra from known samples. The machine learning predictions also require validation with experimental data.
5. The final sentence in the Results and Discussion section, as well as Figure S5, presents a conclusion: "These results show that the prediction of AMP: ADP: ATP = 1:1:1 is highly erroneous in all models. Prediction becomes more difficult when there are more than three molecules with similar structures, such as APs. Therefore, data preprocessing and machine-learning models must be improved to predict more detailed mixing ratios." This implies that using machine learning combined with SERS is unable to simultaneously identify and quantify AMP, ADP, and ATP in a 1:1:1 mixture. Moreover, in actual biological systems, the proportions of AMP, ADP, and ATP typically vary greatly. Can this technique cope with such variability?
6. What are the innovations of this study? There have been many studies combining SERS with machine learning (TrAC Trends in Analytical Chemistry, 2020,124, 115796; Analytical and Bioanalytical Chemistry, 2023, 415, 3945-3966). This research aimed to address the issue of simultaneous detection of AMP, ATP, and ADP, yet the results were not ideal. What is the value of this work?
7. Several methods and conclusions in the paper lack supporting evidence or references. For example, “This is attributed to the electrostatic repulsion caused by the carboxyl groups modified on the nanoparticle surface.” and “The fabrication method selected in this study was a simple and inexpensive method that does not require the expensive equipment used in electron beam lithography and ion etching, which are common fabrication methods for SERS substrates.”
8. Some of the references in the article are outdated.
Author Response
- In section 2.2 SERS Measurements, was a buffer solution used for the preparation of 1 mM AMP? Was the pH adjusted? A suitable pH and solution environment are crucial for maintaining the stability of AMP, ADP, and ATP. This section should provide more specific experimental details.
Reply:
Thank you for your comment. In this experiment, ultrapure water is used as the solvent because solutes in the buffer affect the measurement. Since the mixing solution is used immediately after preparation, we do not expect any change in AMP, ADP, and ATP.
- The statement in Line 190, "Larger nanoparticles have larger areas……from being densely adsorbed" is not entirely accurate. While it is true that larger particles have greater contact areas, which may involve more carboxyl groups participating in electrostatic repulsion, reducing the likelihood of dense adsorption, the repulsion between particles is not solely dependent on the surface contact area. It is also influenced by the uniformity of particle distribution on the substrate, adsorption forces, and solution conditions. Other factors, such as van der Waals forces and geometric constraints, may also play a role in the adsorption process of larger particles. It is recommended to supplement this with more sound physicochemical evidence.
Reply:
Thank you for your advice. As you noted, there could be many other factors. Since nanoparticle adsorption is not the main topic of this study, we do not intend to discuss it in depth. One possible factor was noted in line 190. The text has been slightly modified to avoid misinterpretation.
- This study optimized nanoparticles with diameters of F30 nm, 50 nm, and 100 nm, demonstrating that 100 nm performed best. Could larger diameter nanoparticles potentially exhibit superior performance? It is suggested to test the performance of nanoparticles with larger diameters or provide justification for why larger diameters might be unsuitable.
Reply:
Larger diameter nanoparticles exhibit inferior performance. The absorption peak shifts with nanoparticle size. Therefore, the wavelength of the absorption peak shifts from 785 nm when larger diameter nanoparticles are used. Since the best performance is achieved when the absorption peak is located at approximately 785 nm, we consider nanoparticles with a diameter of 100 nm to be optimal.
- Although the elbow method determined that the number of clusters is four, whether this is truly sufficient to represent the complexity of the data should be further evaluated based on the specific characteristics of the data and experimental results. It is recommended to employ appropriate external validation methods, such as comparison with spectra from known samples. The machine learning predictions also require validation with experimental data.
Reply:
Thank you for your comment. When the number of clusters was increased, the Raman shifts classified in Cluster 4 were subdivided. This subdivision is not meaningful because the Raman shifts in cluster 4 are considered to contain almost no information on chemical structure. Based on the results of the elbow method and this discussion, the number of clusters is set to 4.
In the average spectra in Figure 7, the shapes of the peaks due to the chemical structure are different. Since these peaks are extracted by feature selection and used for training, the predictions of machine-learning model are based on the chemical information contained in the spectra.
- The final sentence in the Results and Discussion section, as well as Figure S5, presents a conclusion: "These results show that the prediction of AMP: ADP: ATP = 1:1:1 is highly erroneous in all models. Prediction becomes more difficult when there are more than three molecules with similar structures, such as APs. Therefore, data preprocessing and machine-learning models must be improved to predict more detailed mixing ratios." This implies that using machine learning combined with SERS is unable to simultaneously identify and quantify AMP, ADP, and ATP in a 1:1:1 mixture. Moreover, in actual biological systems, the proportions of AMP, ADP, and ATP typically vary greatly. Can this technique cope with such variability?
Reply:
Thank you for your comment. With this technique, it is difficult to apply the technique to biological samples that contain impurity. However, this study has shown that even with a small amount of data, it is possible to classify mixing ratios with an accuracy of about 0.9. In the future, it is expected that classification of biological samples will become possible by increasing the amount of measurement data with different mixing ratios and by incorporating deep learning models that can perform complex analysis.
- What are the innovations of this study? There have been many studies combining SERS with machine learning (TrAC Trends in Analytical Chemistry, 2020,124, 115796; Analytical and Bioanalytical Chemistry, 2023, 415, 3945-3966). This research aimed to address the issue of simultaneous detection of AMP, ATP, and ADP, yet the results were not ideal. What is the value of this work?
Reply:
The value of this paper is that it incorporates feature selection and data augmentation into the analysis of SERS spectra to improve prediction accuracy for small amounts of data. Although feature selection and data augmentation are commonly used in image recognition, there are few examples of its application to the SERS spectra analysis. In this study, we incorporated these techniques and confirmed that prediction accuracy was improved. Simultaneous recognition and detection of APs, which are similarly structured molecules, was also achieved with an accuracy of approximately 0.9. Although the accuracy was not ideal, we think that we have contributed to the future development of combined SERS and machine learning analysis.
- Several methods and conclusions in the paper lack supporting evidence or references. For example, “This is attributed to the electrostatic repulsion caused by the carboxyl groups modified on the nanoparticle surface.” and “The fabrication method selected in this study was a simple and inexpensive method that does not require the expensive equipment used in electron beam lithography and ion etching, which are common fabrication methods for SERS substrates.”
Reply:
Thank you for your advice. References have been added to the above two sentences.
- Some of the references in the article are outdated.
Thank you for your advice. Some references have been updated.
Reviewer 2 Report
Comments and Suggestions for Authors
1) English should be revised.
2) There's an error in the caption of Figure 4 or in the text in the figure itself. The caption indicates that (a) is the subfigure relative to GPs, and (b) is relative to GNDAs; the figure has text indicating the opposite. Revise the main text accordingly if necessary.
3) In table S1, the full names of the models would be useful as many readers might not be familiar with all of them. Furthermore, the actual name of the function should be added somewhere including the full subpackage(s) (e.g. "sklearn.neighbors.KNeighborsClassifier"), as the same package can contain different implementations of the same algorithm or slightly different algorithms commonly referred to with the same name (e.g. sklearn.cross_decomposition.PLSCanonical VS sklearn.cross_decomposition.PLSRegression).
4) Line 180: the authors were thorough in their declaring of the Python version, however this is inconsequential to the paper since they are all functionally the same. It would be wise for future-proofing the paper that they declared in line 175 the scikit-learn version instead, since the algorithms implementations and names can change from version to version.
5) Too little information is given for each classification technique. Critical information on these is missing from the main text and from the supplementary information, and the information given is not commented at all. Some egregious examples of this is the lack of any information on the SVM kernel function, the midpoint of the logistic regression curve, LDA's solver and possible shrinkage. It appears to be implied that the authors employed the default scikit-learn values for these parameters, but knowledge of some of these is essential for anyone trying to replicate or emulate the study with tools different than scikit-learn, and for general understanding of the underlying mechanisms of the algorithms.
6) Equation 1 states that the standardization utilized for mean centering the "I" value as the "average intensity of the entire spectrum" and "s" "is the standard deviation of the intensity of the entire spectrum" [sic]. In machine learning and chemometrics, this sort of operation is carried out "vertically" instead of "horizontally" as the authors did, i.e. the average and standard deviation used for mean centering and standardization of each Raman shift are not those of one entire spectrum, but those of the group composed by each Raman shift of every spectrum (i.e.: to standardize the intensity of the first Raman shift of each spectrum, "I" and "s" should be the average and standard deviation of the first Raman shift of all spectra instead). The operation declared by the authors is more akin to a spectral normalization, which is usually carried out separately from machine learning standardization and methods such as l1, l2, or dividing by the intensity of the highest signal in the spectrum are used instead.
7) Lines 228-242: the FDTD method calculation is a very interesting addition, and pertinent with the topic. However: a) Critical information about this part is missing. How was this calculated? Which software was used? No such information is given in Materials and Methods, but this and more details are needed. b) What are the authors trying to predict or demonstrate with this? What new information is given and how does this help? What conclusions can we evince from this? No mention in the introduction and conclusion was written about this. The authors should try to insert these FDTD simulations more organically in the text.
However, the following are the most critical points to be addressed.
8) Lines 250-259: the authors assign the bands but do not attempt to give any explanation on why they should differ so much among the three molecules. For example, the ring breathing mode at 730 cm-1 and the 1460 cm-1 band vary a lot among the spectra, but the molecule in that region is essentially the same. One would expect the P=O and -OH bands to bear the most difference since the amount of substance is much larger among 1:0:0 and 0:0:1, for instance. Was this expected from the authors? What were the expected and unexpected outcomes of the experiments, from a spectral point of view?
9) Furthermore, no spectrum of the pure substances with non-enhanced Raman is given. This would be the very first step in any study of this kind. The authors should lead with non-enhanced Raman spectra on these molecules (pure), then explain whether differences can be seen with respect to SERS or not. Moreover, SERS spectra usually are selectively enhanced in specific regions because of the plasmon resonance radiation polarization: does this happen? Are SERS spectra similar within group, and what differences can be observed? These considerations are of interest for the scientific community, but they are also important for this sort of investigation for data verification in SERS.
Given the very few data points and the data augmentation techniques employed in the study, points (8) and (9) of this review are even more critical.
10) Very little information on the outcomes of each of the classification techniques is given (e.g. no scatterplot is given, no decision boundaries is drawn, no information on the final models seven classification algorithms employed except confusion matrices for only three of them). More information is much needed to understand the value of the proposed approach in each model and to evaluate the quality of the models.
11) No cross validation was attempted. These are very few data points for training, especially since the dimensionality of the dataset is still extremely high even after excluding cluster 4 (269 features and 5 data points per group for a grand total of 35 data points for training). Consequently, the choice of the spectra in the training dataset VS the test dataset is crucial. How does the choice of spectra in the training dataset actually influence the outcome? Some sort of cross validation, e.g. iteratively choosing different spectra for training, should be attempted to understand just how robust the study is to different choices of spectra for the training dataset.
12) The purpose of the study was to demonstrate the possibility to correctly classify unknown samples with SERS into seven different classes. However, the classes here are heavily correlated, because three classes are pure substances, three are binary mixtures of two out of the three pure substances, and one is the ternary mixture. From an applicative point of view, how is dividing unknown spectra into 1:0:0, 1:1:0 and 1:1:1 classes more advantageous than trying relative quantification by regression? In real-life samples, one would expect the ratios to vary continously. Was relative quantification attempted? Or is there any reason why this classification is better suited for the task? The authors should either investigate relative quantification or justify their approach in the scope of the paper.
English is understandable and adequate, except for some minor revisions needed due to incorrect form
Author Response
- English should be revised.
Reply:
Our manuscript has been edited by "Editage (professional editing company)".
- There's an error in the caption of Figure 4 or in the text in the figure itself. The caption indicates that (a) is the subfigure relative to GPs, and (b) is relative to GNDAs; the figure has text indicating the opposite. Revise the main text accordingly if necessary.
Reply:
Thank you for your comment. The text has been revised.
- In table S1, the full names of the models would be useful as many readers might not be familiar with all of them. Furthermore, the actual name of the function should be added somewhere including the full subpackage(s) (e.g. "sklearn.neighbors.KNeighborsClassifier"), as the same package can contain different implementations of the same algorithm or slightly different algorithms commonly referred to with the same name (e.g. sklearn.cross_decomposition.PLSCanonical VS sklearn.cross_decomposition.PLSRegression).
Reply:
Thank you for your advice. We have revised table S1.
- Line 180: the authors were thorough in their declaring of the Python version, however this is inconsequential to the paper since they are all functionally the same. It would be wise for future-proofing the paper that they declared in line 175 the scikit-learn version instead, since the algorithms implementations and names can change from version to version.
Reply:
Thank you for your advice. We have added version information of scikit-learn.
- Too little information is given for each classification technique. Critical information on these is missing from the main text and from the supplementary information, and the information given is not commented at all. Some egregious examples of this is the lack of any information on the SVM kernel function, the midpoint of the logistic regression curve, LDA's solver and possible shrinkage. It appears to be implied that the authors employed the default scikit-learn values for these parameters, but knowledge of some of these is essential for anyone trying to replicate or emulate the study with tools different than scikit-learn, and for general understanding of the underlying mechanisms of the algorithms.
Reply:
Thank you for your comment. As noted in the manuscript (Sec. 2.4), parameters other than those shown in Table S1 are default values in scikit-learn. Since these values can be found on the official scikit-learn page, we consider it unnecessary to include these values in the text.
scikit-learn: machine learning in Python — scikit-learn 1.4.2 documentation
- Equation 1 states that the standardization utilized for mean centering the "I" value as the "average intensity of the entire spectrum" and "s" "is the standard deviation of the intensity of the entire spectrum" [sic]. In machine learning and chemometrics, this sort of operation is carried out "vertically" instead of "horizontally" as the authors did, i.e. the average and standard deviation used for mean centering and standardization of each Raman shift are not those of one entire spectrum, but those of the group composed by each Raman shift of every spectrum (i.e.: to standardize the intensity of the first Raman shift of each spectrum, "I" and "s" should be the average and standard deviation of the first Raman shift of all spectra instead). The operation declared by the authors is more akin to a spectral normalization, which is usually carried out separately from machine learning standardization and methods such as l1, l2, or dividing by the intensity of the highest signal in the spectrum are used instead.
Reply:
Thank you for your comment. As you pointed out, we used the standardization as the normalization. Although this process is not often used in machine learning, it was employed because of its higher prediction accuracy than the normalization.
- Lines 228-242: the FDTD method calculation is a very interesting addition, and pertinent with the topic. However: a) Critical information about this part is missing. How was this calculated? Which software was used? No such information is given in Materials and Methods, but this and more details are needed. b) What are the authors trying to predict or demonstrate with this? What new information is given and how does this help? What conclusions can we evince from this? No mention in the introduction and conclusion was written about this. The authors should try to insert these FDTD simulations more organically in the text. However, the following are the most critical points to be addressed.
Reply:
Thank you for your comment. The FDTD results are an auxiliary addition; this was used to confirm the electric field enhancement. Since this result is not the main topic of this study, it was deemed unnecessary in the introduction and conclusions. We have added the name of the software used for the simulations and information on the models created.
- Lines 250-259: the authors assign the bands but do not attempt to give any explanation on why they should differ so much among the three molecules. For example, the ring breathing mode at 730 cm-1 and the 1460 cm-1 band vary a lot among the spectra, but the molecule in that region is essentially the same. One would expect the P=O and -OH bands to bear the most difference since the amount of substance is much larger among 1:0:0 and 0:0:1, for instance. Was this expected from the authors? What were the expected and unexpected outcomes of the experiments, from a spectral point of view?
Reply:
Thank you for your comment. The large difference in the 730 cm-1 and 1460 cm-1 bands in each sample was unexpected. Since it has been reported that APs interact with the gold surface (J. Phys. Chem. C 2009, 113, 14390–14397), we think that this interaction changes the state of the APs. This interaction depends on the number of phosphate groups, and we assume that this difference affects the shape of the SERS spectra. The details of the mechanism need to be investigated by quantum chemical calculations such as Gaussian. Since it is not the main topic of this paper, it is not mentioned.
- Furthermore, no spectrum of the pure substances with non-enhanced Raman is given. This would be the very first step in any study of this kind. The authors should lead with non-enhanced Raman spectra on these molecules (pure), then explain whether differences can be seen with respect to SERS or not. Moreover, SERS spectra usually are selectively enhanced in specific regions because of the plasmon resonance radiation polarization: does this happen? Are SERS spectra similar within group, and what differences can be observed? These considerations are of interest for the scientific community, but they are also important for this sort of investigation for data verification in SERS.
Reply:
ADP and ATP are very hygroscopic. Because they quickly absorb moisture, normal Raman measurements were not stable. Therefore, comparison with SERS spectra was difficult and was omitted in this paper. Since we think that non-enhanced Raman of APs will be necessary if the attribution of the spectra and the mechanism of detection are pursued in detail. This is an issue for future study.
- Very little information on the outcomes of each of the classification techniques is given (e.g. no scatterplot is given, no decision boundaries is drawn, no information on the final models seven classification algorithms employed except confusion matrices for only three of them). More information is much needed to understand the value of the proposed approach in each model and to evaluate the quality of the models.
Reply:
Thank you for your comment. Adding scatter plots and boundary conditions focuses the discussion on the model. However, what we would like to discuss here is not the model but its predicted results. For this reason, we did not include these details. Then, only the confusion matrix of the model with the highest prediction accuracy was shown. The models with lower accuracy were not mentioned in the paper, so we determined that it was not necessary to show the confusion matrix.
- No cross validation was attempted. These are very few data points for training, especially since the dimensionality of the dataset is still extremely high even after excluding cluster 4 (269 features and 5 data points per group for a grand total of 35 data points for training). Consequently, the choice of the spectra in the training dataset VS the test dataset is crucial. How does the choice of spectra in the training dataset actually influence the outcome? Some sort of cross validation, e.g. iteratively choosing different spectra for training, should be attempted to understand just how robust the study is to different choices of spectra for the training dataset.
Reply:
Thank you for your comment. We have confirmed that the choice of data for training can change the prediction results. However, the variation was small. In this study, no cross-validation was attempted because the focus was on feature selection and data augmentation to improve prediction accuracy.
- The purpose of the study was to demonstrate the possibility to correctly classify unknown samples with SERS into seven different classes. However, the classes here are heavily correlated, because three classes are pure substances, three are binary mixtures of two out of the three pure substances, and one is the ternary mixture. From an applicative point of view, how is dividing unknown spectra into 1:0:0, 1:1:0 and 1:1:1 classes more advantageous than trying relative quantification by regression? In real-life samples, one would expect the ratios to vary continously. Was relative quantification attempted? Or is there any reason why this classification is better suited for the task? The authors should either investigate relative quantification or justify their approach in the scope of the paper.
Reply:
Thank you for your comment. Since this study is a demonstration, it was tested with simple classifications. Therefore, in this study, we selected these seven different classes. Since the ratio of each APs is a biologically important factor, it is important to predict the ratio. In the future, we will increase the variation of mixing ratios in the training data so that we can predict the ratio of each APs rather than class classification.
Round 2
Reviewer 1 Report
Comments and Suggestions for Authors
I appreciate the authors' efforts in addressing the concerns raised during the first review of this manuscript. The revisions made have significantly improved the clarity and overall quality of the work.
The authors have successfully addressed all major issues identified in the initial review. The revisions are well-articulated and enhance the manuscript’s scientific rigor.
The discussion now provides a more comprehensive interpretation of the findings, situating them within the broader context of combined SERS and machine learning analysis.
Overall, the manuscript is now suitable for publication. I recommend that it be accepted in its current form.
Thank you for the opportunity to review this revised manuscript.
Author Response
Thank you very much for reviewing our manuscript and for your valuable comments.

Reviewer 2 Report
Comments and Suggestions for Authors
The authors answered some of the issues presented to them in this review. The responses were appreciated. Nonetheless, only the least important points were addressed.
Most of the critical points (the last ones) were left unanswered in the manuscript. The findings might be generated by improper preprocessing, low initial sampling not taking into account all variability (including SERS variability), choice of hyperparameters in models, choice of train/test data splitting, and more.
In the present state, this reviewer finds that the study could hardly be reproduced by peers, and that the main claim of the manuscript is not demonstrated. However, it can be if at least most of the critical points (points 9 to 12) were addressed.
Comments on the Quality of English LanguageThe authors claim to have employed an editing service. It appears there still are some inaccuracies in the manuscript.
Author Response
Reviewer 2
The authors answered some of the issues presented to them in this review. The responses were appreciated. Nonetheless, only the least important points were addressed.
Most of the critical points (the last ones) were left unanswered in the manuscript. The findings might be generated by improper preprocessing, low initial sampling not taking into account all variability (including SERS variability), choice of hyperparameters in models, choice of train/test data splitting, and more.
In the present state, this reviewer finds that the study could hardly be reproduced by peers, and that the main claim of the manuscript is not demonstrated. However, it can be if at least most of the critical points (points 9 to 12) were addressed.
- Furthermore, no spectrum of the pure substances with non-enhanced Raman is given. This would be the very first step in any study of this kind. The authors should lead with non-enhanced Raman spectra on these molecules (pure), then explain whether differences can be seen with respect to SERS or not. Moreover, SERS spectra usually are selectively enhanced in specific regions because of the plasmon resonance radiation polarization: does this happen? Are SERS spectra similar within group, and what differences can be observed? These considerations are of interest for the scientific community, but they are also important for this sort of investigation for data verification in SERS.
Reply:
Conventional Raman spectra of AMP, ADP and ATP in powder form were obtained. Then, they were compared with the SERS spectra. The results showed that the bands around 730, 1330 and 1460 cm-1 were significantly enhanced. It was also observed that the degree of enhancement of each band differed depending on the analytes (AMP, ADP, or ATP). This is thought to be because the number of phosphate groups changes the state, position, and direction of adsorption. We think that molecular chemistry calculations and other experiments are needed to prove and to provide a deeper insight into this. However, since this is not the main topic of this paper, we do not consider it necessary to pursue it further. These were also added in this paper.
- Very little information on the outcomes of each of the classification techniques is given (e.g. no scatterplot is given, no decision boundaries is drawn, no information on the final models seven classification algorithms employed except confusion matrices for only three of them). More information is much needed to understand the value of the proposed approach in each model and to evaluate the quality of the models.
Reply:
Since there are seven models and three types of training data (RD, FSD, and DAD) in this study, there are 21 outcomes. Since showing and discussing all these results would be very verbose, we decided that it would not be appropriate as a “Communication”. Therefore, only the confusion matrix was added as Figure S5. We have also mentioned a few of these in the manuscript.
- No cross validation was attempted. These are very few data points for training, especially since the dimensionality of the dataset is still extremely high even after excluding cluster 4 (269 features and 5 data points per group for a grand total of 35 data points for training). Consequently, the choice of the spectra in the training dataset VS the test dataset is crucial. How does the choice of spectra in the training dataset actually influence the outcome? Some sort of cross validation, e.g. iteratively choosing different spectra for training, should be attempted to understand just how robust the study is to different choices of spectra for the training dataset.
Reply:
A method using cross-validation was employed to verify accuracy. The data was divided into two parts, one for training and the other for validation. Patterns in which training and validation were interchanged were also conducted. The discussion was revised because the results changed slightly due to a change in the validation methodology.
- The purpose of the study was to demonstrate the possibility to correctly classify unknown samples with SERS into seven different classes. However, the classes here are heavily correlated, because three classes are pure substances, three are binary mixtures of two out of the three pure substances, and one is the ternary mixture. From an applicative point of view, how is dividing unknown spectra into 1:0:0, 1:1:0 and 1:1:1 classes more advantageous than trying relative quantification by regression? In real-life samples, one would expect the ratios to vary continously. Was relative quantification attempted? Or is there any reason why this classification is better suited for the task? The authors should either investigate relative quantification or justify their approach in the scope of the paper.
Reply:
Since the ratio of each APs is a biologically important factor, it is important to predict the ratio. However, there is no report that has been able to determine the ratio of three or more APs using SERS. Therefore, in this study, we tested a simple class classification, seven classifications (1:0:0, 0:1:0, 0:0:1, 1:1:0, 1:0:1, 0:1:1, and 1:1:1), as a demonstration. Since this study focuses on limited data analysis using feature selection and data augmentation, detailed ratio measurements were not performed. Based on the results of this study, we will increase the variation of mixing ratios in the training data so that we can predict the ratio of each APs rather than class classification.
Round 3
Reviewer 2 Report
Comments and Suggestions for Authors
The paper has been greatly improved in public usability after these revision, in this reviewer's opinion, the changes in training/test data numbers makes this way better than before. I hope that the training/test data was acquired in a way to test robustness.